



# Analytical Study of North Indian Oceanic Cyclonic Disturbances with Special Reference to Extremely Severe Cyclonic Storm Fani: Meteorological Variability, India's Preparedness with Terrible Aftermath

Soumen Chatterjee[1]

[1] Department of Geography, University of Burdwan, Golapbag, Purba Bardhaman 713 104, West Bengal, India.

*Correspondence to*: Soumen Chatterjee (geosoumchat@gmail.com)

**Abstract.** Having a total coastal tract of about 7,516 km with 5,400 km long mainland coastline, India is highly vulnerable to natural hazards like tropical cyclones (TCs). The analysis based on the historical dataset (1891-2019) of TCs over North

Indian Ocean (NIO) also claims that the four coastal states (Andhra Pradesh, Odisha, Tamil Nadu and West Bengal) and one union territory (Pondicherry) on the east coast frequently face cyclonic storm than other coastal parts of India. The seasonal distribution (Pre-monsoon, Monsoon and Post-monsoon) of cyclonic storms over the Arabian Sea (AS) and Bay of Bengal (BoB) in last 150 years also help to unfold the fact that the Odisha and West Bengal coast are exposed to TCs mostly during the monsoon season (June to September) encompassing with strong winds, heavy rainfall and high storm surge. The

extremely severe cyclonic storm (ESCS) Fani is the rarest summer cyclones, the first one in 43 years to strike the coastal part of Odisha on May 3, 2019 and one of the three worst cyclones in last 150 years with a sustained surface wind speed of 175-180 kmph. Odisha has been affected horribly due to the vulnerability of Fani. Although the death toll was limited within 64 due to rapid evacuation of nearly 1.68 million people, the killer cyclone has caused irreparable damages in social sectors (housing, education and food security), productive sectors (agriculture, fisheries and livestock) and also informative sectors

(power, telecommunication, road, water facilities and public buildings). The estimated costs have reached nearly 4.18 billion USD only in Odisha. The southern part of West Bengal has also affected badly due to intense downpour and very high storm surges (2-3 m above mean sea level). To map the flooded areas of Odisha and West Bengal due to intense rainfall (cause inland flooding) and storm surges (cause coastal flooding), the Sentinel-1 SAR GRD dataset has also been used in Google Earth Engine (GEE) environment to link with the deadly cyclone Fani. So, the present study successfully advocates the

historical background of TCs over NIO with particular reference to ESCS Fani including its meteorological variability, preparedness and the trail of devastation.



## 1 Introduction

Tropical cyclones (TCs), one of the biggest threats to life and property have caused 0.77 million death and 1,407.6 billion USD economic loss worldwide in last fifty years with an average of 43 deaths and 78 million USD damages every day
(WMO, 2020). India is not far behind in this regard for having about 5,400 km long mainland coastline out of total 7,516 km coastal tract (Mohapatra, 2015; Mohapatra et al., 2012; NCRMP, 2019). So, the country is also highly vulnerable to the natural hazard like TCs which are originated in the North Indian Ocean (the Arabian Sea and Bay of Bengal), although the region generates only 7 % of the world's TC (Dube et al. 1997; Mohapatra, 2015). Out of 96 cyclone-affected districts of India (including 72 coastal districts), 12, 41, 30 and 13 districts come under very high, high, moderate and less cyclone
hazard-prone categories, respectively (Mohapatra et al., 2012; RSMC, 2018a), based on the frequency of total cyclones and severe cyclones, actual/estimated maximum wind strength, Probable Maximum Storm Surge (PMSS) and Probable Maximum Precipitation (PMP). Besides, the studies of Dube et al. (1997), Mohapatra (2015), Mohapatra et al. (2012) and RSMC (2018) also claim that almost all the coastal districts of West Bengal, Odisha, Andhra Pradesh and Tamil Nadu come under high to very high cyclone hazard-prone category because the majority of cyclones occur in Bay of Bengal (BoB) as
compared to the Arabian Sea (AS). On the other hand, almost the entire eastern coastal tract of India has been identified as very high damage risk zone in terms of wind hazard vulnerability of India by the Building Materials & Technology Promotion Council (BMTPC, 2019) under Ministry of Housing & Urban Affairs, Govt. of India.

The year, 2019 has experienced four very severe or more intense cyclonic storms, but the Extremely Severe Cyclonic Storm (ESCS) Fani is the only tropical cyclone of the year that has made landfall as a very severe cyclonic storm in the coastal part
of Odisha (Puri district) on 3 May (RSMC, 2020). It is also the strongest TC to strike Odisha coast since 1999 super cyclone not only in terms of intensity but also from the perspective of damage in social sectors (housing, education and food security), productive sectors (agriculture, fisheries and livestock) and also informative sectors (power, telecommunication, road, water facilities and public buildings). Despite taking several pre-storm emergency preparedness plans, ESCS Fani has succeeded to cause large damage of an estimated cost of 4.18 billion USD only within Odisha (India) with 64 human
casualty (OSDMA, UN India, WB and ADB, 2019; UNICEF, 2019). So, the present study has focused on the detailed scenario of cyclonic storm distribution over North Indian Ocean (NIO) and trend analysis from 1891 to 2018 with particular reference to Fani and its devastating impact on lives, property and economy due to high-speed winds, excessive rainfall, storm surge and flooding.

## 2 Historical background of cyclonic disturbances

The states like Andhra Pradesh, Odisha and West Bengal situated along the eastern coast of India have experienced severe damages due to TCs originated in the BoB (Chittibabu et al., 2004; Dube et al. 1997; Dube et al. 2000a, 2000b; Mohapatra, 2015; Mohapatra et al., 2012; Rao et al., 2006) because the most of the cyclonic storms tend to occur in the BoB basin as compared to the AS basin with an approximate ratio of 3:1. Naturally, the prepared gridded image (Figure 1a) based on the





genesis of cyclonic storms data over NIO basin by Indian Meteorological Department (IMD) indicates that the BoB has

experienced very high frequency of cyclonic storms cumulatively since 1891. On the other hand, the inter-annual average values of the past 128 years (1891-2019) also specify that the BoB experiences almost 2 severe or more intense cyclonic storms out of every 6 TCs, whereas the AS originates almost 1 severe or more intense cyclonic storms out of every 2 TCs. So, the above discussion helps to understand the reason behind belonging of the coastal tract of northern Bay of Bengal to the very highly vulnerable cyclone hazard-prone zone with having potentiality to generate high storm surge and killer strong

gusty winds. The further analysis (Figure 1b) also explains that the most of the TCs in the lower latitudes in association with Inter-tropical Convergence Zone (ITCZ) occur during pre-monsoon (March-May) and post-monsoon (October – December) seasons, but the coastal states along with the northern BoB and AS have faced most the cyclonic storms during monsoon season (June – September). So, the high-intensity devastating summer cyclone is the rarest event in the long cyclonic history of northern BoB.

Extensive work has been also done by Mooley (1980), Singh et al. (2000, 2001), Singh (2007), Srivastava et al. (2000) and Mohapatra et al. (2017) for determining the change analysis in the intensity and frequency distribution of TCs over BoB and AS. In the present study, the temporal frequency distribution data (1891 - 2019) of cyclonic disturbances by IMD (RSMC, 2019) has been analyzed for obtaining long-term trend of TCs over NIO. The analysis also helps to unfold the fact that the 30 years moving average lines of overall cyclonic storms and severe or more intense TCs almost follow the constructed

regression lines for the period of 1891 to 2019 (Figure 2a). To analyze the historical trend of TCs in more detail, the intensification rate of cyclonic storms to severe or more intense cyclonic storms has been examined over BoB and AS for the past 128 years. The results (trend lines in Figure 2b & 2c) help to understand that the changes in the intensification rate of cyclonic storms to severe or more intense TCs have increased over time in case of both the BoB and AS with the $R^2$ values of 0.0985 and 0.0054, respectively.

The coastal districts of Odisha have experienced several cyclonic storms with strong winds, floodings and very high storm surges, but Fani is the rarest summer cyclones, the first in 43 years and one of the three worst cyclones in last 150 years to strike the coastal part of Odisha with immense economic and social impact (OSDMA, UN India, WB and ADB, 2019; UNICEF, 2019). Then it further proceeds towards West Bengal and finally weakens over Bangladesh and its adjoining Indian territory (Central Assam).

**3 Data used and methods**

Large numerical datasets, pre-storm & post-storm images and extensive study are required for analyzing the historical background of TCs over NIO, the meteorological overview of ESCS Fani (wind speed data, atmospheric pressure data, rainfall details & storm surge information) and damage details including inundation statistics with causality due to the devastation caused by the cyclone Fani. The data related to the frequencies of TCs over the BoB and the AS for the period of

1891 to 2019 have been obtained from the website of RSMC (2019) for Tropical Cyclones over North Indian Ocean in



association with IMD. Two types of images (Table 1) have been used for the analysis and visualization of the different aspects of the ESCS Fani. Synthetic Aperture Radar (SAR) images of Sentinel-1 (pre-storm and post-storm) have been examined in Google Earth Engine (GEE) environment, a cloud-based platform for planetary-scale environmental data analysis (Gorelick et al., 2017) for inland and coastal inundation mapping due to heavy rainfall and storm surges during Fani

by dividing the entire affected area of West Bengal and Odisha into two windows (Figure 3). The first window includes the southern districts of West Bengal and its surroundings. On the other hand, the second window includes the eastern portion of Odisha and surroundings. The flooded areas have been mapped by using certain codes in GEE. For the further analysis, the reliability of extracted flooded areas has been measured by collecting 80 samples for each classification unit (viz., water bodies, flooded lands and non-flooded lands) from both windows which have been referred as the region of interest (ROI) in

Figure 3. The accuracy assessment table (Table 2) indicates that the computed overall accuracy values are very high in case of VV and VH polarization for both windows. Besides, GPM IMERG Final Precipitation (0.1 degree x 0.1 degree, V06) data (Huffman, 2019) have also been used for the analysis of daily rainfall intensity during the storm event in Giovanni environment. The track related information and other meteorological details regarding cyclonic storm Fani have been collected from the archive section of the RSMC for Tropical Cyclones over North Indian Ocean website

(http://www.rsmcnewdelhi.imd.gov.in/index.php?lang=en) as in the form of 'Best Tracks Data' and 'Bulletins', respectively. The storm surge height information along the coastal tract of Odisha and West Bengal has also been collected from the Emergency Response Coordination Centre (ERCC, 2019) portal in association with Joint Research Centre (JRC) under Hurricane Weather Research and Forecasting (HWRF) of NOAA.

Very few research works have been done by Acharyya et al. (2020), Kumar et al. (2020), Liu and Zhu (2020), and Mohanty

et al. (2020a, 2020b) on different aspects of cyclonic storms Fani. Several reports have also been published based on the impact assessment of ESCS Fani by different reliable sources like UNICEF (2019), OSDMA, UN, WB & ADB (2019), IRC (2019) and Humanity Road (2019). The summary of the research articles and published reports have been enlisted in Table 3. Based on the available information, it can be claimed that the present study has its significance in terms of subject matter because no discussion has been made so far in any study where the coastal and inland flooded zones have been delineated

with the calculation of inundation statistics due to heavy rainfall and storm surge during cyclone Fani. Besides, this is the only article where India's preparedness and damage statistics have also discussed briefly in the context of cyclonic storm Fani with the long historical background of TCs over the NIO.

## 4 Synoptic history of ESCS Fani

The cyclonic storm Fani has covered long distance (nearly 3090 km) combinedly over the Bay of Bengal, Indian and

Bangladesh landmass (Figure 4). It has spawned as a depression (D) over the eastern equatorial Indian Ocean & adjoining southeast BoB on April 26, 2019. Then it has moved northwestwards with a speed of about 56 kmph and has intensified into a deep depression (DD) over southeastern BoB on April 27. It has again moved north-northwestwards and has intensified





into Cyclonic Storm (CS) at 11:30 hrs IST on the same date near latitude 5.2 ° N and longitude 88.6 ° E. After intensifying into CS, it has again moved northwestwards and has laid centred near latitude 5.4°N and longitude 88.5°E at 14:30 hrs IST. At about 17:30 hrs IST on that very day, it has curved towards the north. After gaining more speed, Fani has further intensified into a Severe Cyclonic Storm (SCS) near 10.1°N latitude and 86.7°E longitude over the central portion of the northern BoB at 17:30 hrs IST on April 29. This rarest summer cyclone has transformed into a Very Severe Cyclonic Storm (VSCS) at 05:30 hrs IST on the very next day. After moving northwestwards, it has further intensified into an Extremely Severe Cyclonic Storm (ESCS) on April 30 at 17:30 hrs IST. Although the cyclonic storm Fani has continued its journey as ESCS till 08:30 hrs IST of May 3, it has geared up its speed initially up to almost 213 kmph with gusting to 210 kmph till May 2 (at about 20:30 hrs IST) and has weakened a little bit on the very next day. The ESCS Fani has made landfall near Puri (Odisha Coast) as VSCS with maximum sustained wind speed of nearly 157 kmph with gusting to 205 kmph between 08:00 to 10:00 hrs IST on May 3, 2019. After crossing the Odisha coast, it has further continued to move north-northeastwards and has weakened gradually after emerging into Gangetic West Bengal as an SCS with the wind speed of 111 Kmph with gusting to 115 Kmph by the early morning of May 4. The remnants of SCS Fani has moved towards east-northeastwards and has weakened into a DD over Bangladesh. Finally, it has transformed into a well-marked low pressure over central Assam & neighbourhood at 23:30 hrs IST on May 4, 2019. The best track details of ESCS Fani have been presented in Table 4.

The meteorological observations include the sustained wind speed and also the atmospheric pressure (Figure 5) during the storm event. Fani's maximum wind speed of nearly 213 kmph has occurred over NW BoB just fifteen hours before of making landfall on the Odisha coast at 20:30 hrs IST on May 2. The minimum pressure of cyclonic storm Fani has been estimated to be 932 hPa on that particular day. On the other hand, the maximum wind speed has observed nearly 157 kmph with 966 hPa atmospheric pressure (the lowest estimated central pressure after making landfall) at the time of striking Puri, one of the coastal districts of Odisha at 11:30 hrs IST on May 3. The correlation coefficient value (r) is indicating the strong negative relationship (- 0.98) between two weather variables (wind speed and atmospheric pressure) which are collected along the best track of summer cyclonic storm Fani.

## 5 Results and Discussion

This particular section of the study includes three main aspects which are intimately related to cyclonic storm Fani, viz., India's preparedness, inundation details and other damage statistics. The first one (disaster preparedness) helps to understand how India has abled to tackle and overcome the situation by taking management proper steps and keeping the death toll in control. Last two aspects (inundation details with other damage statistics) help to get the idea about the devastation caused by ESCS Fani, in spite of taking the necessary steps to handle the cyclone situation.



## 5.1 India's preparedness

The pre-storm and post-storm disaster preparedness programmes always create several challenging situations for the country

like India, the world's second-most populous land with 1,366 million residents (United Nations, 2019). In case of the tropical cyclone, the coastal districts of India have to face more critical situations for having high population density. Nearly 40% of the total population is more or less exposed to cyclones because of living within 100 km of coastline in India (NCRMP, 2019). The strong winds with heavy rainfall and storm surge during cyclonic storms can cause severe damage that includes complete or partial destruction of homes, buildings, roads, power connections and water outages (FEMA, 2017). So, the

proper pre-storm emergency preparedness plan takes an important role to reduce the damages and casualties caused by an unavoidable disaster like the tropical cyclone.

The National Disaster Response Force (NDRF), Odisha State Disaster Management Authority (OSDMA), West Bengal Disaster Management & Civil Defence Department (WBDM & CD), Andhra Pradesh State Disaster Management Authority (APSDMA), Indian Red Cross Society and United Nations (UN) have worked joint handedly for ensuring safety to the

common people who are living along the track of ESCS Fani. Nearly 1.68 million persons have been evacuated from the path of Fani of three affected states (Odisha, West Bengal and Andhra Pradesh) of India (UNICEF, 2019). This is considered as the largest evacuation process not only in the cyclonic history of India but also in the cyclonic history of the World. A large number of cyclone or flood relief camps (9000 in Odisha, 471 in West Bengal and 120 in Andhra Pradesh) have been used temporarily for the relocation of evacuated people during the storm event. More than 45,000 volunteers, 2,000

emergency workers, 100,000 officials have worked day and night in the rescue operation in Odisha to fight against this deadly tropical cyclone. A combined amount of nearly 153 million USD has been released in advance from State Disaster Response Fund (SDRF) to State Governments of Odisha, West Bengal and Andhra Pradesh. As Fani has made landfall near Puri of Odisha Coast, special attention has been paid to the health sector as pre and post-storm measures by Odisha government ((IRC, 2019). After the storm event, 302 affected public health centres have been restored for providing

emergency treatment. 184 mobile medical teams have been deployed for minor injured persons. 1,945 pregnant women have also been shifted to delivery points. Besides, several other measures have also been taken by the national, state and local government to battle against this monstrous extremely severe cyclonic storm. IMD received appreciation from the World Meteorological Organization (WMO) and other national & international scientific community and media for pinpoint accuracy during the storm event. The United Nations Office for Disaster Risk Reduction (UNDRR) has also praised India's

zero casualty approach to manage extreme weather events like Fani.

## 5.2 Inundation detail

Effective monitoring of floods is quite impossible without the use of satellite images. GEE has made the work easier than before as because of its capabilities to detect temporal changes on the earth's surface (Uddin et al., 2019; DeVries et al., 2020; Agnihotri et al., 2019; Clement et al., 2017; Kussul et al., 2011) using the Sentinel-1's C-band SAR active sensor





derived images, an independent dataset of any time of the day or night, regardless of weather and environmental conditions
(ESA, 2020). So this particular cloud-based platform has been used for monitoring flood situation due to heavy rainfall
(responsible for inland flooding) and storm surge (caused coastal inundation) during ESCS Fani.

The rainfall associated with ESCS Fani based on daily accumulated precipitation extracted from GPM IMERG Final
Precipitation (0.1 degree x 0.1 degree, V06) data in Giovanni environment has been represented in Figure 6 from April 26 to

May 4, 2019. It indicates the occurrence of heavy to very heavy downpour over the coastal Odisha, Gangetic West Bengal
and adjoining Bangladesh on May 3. Most of the places have experienced more than 75 mm accumulated rainfall in 24 hrs
on that very day. As the cyclonic storm Fani has moved towards northeastern India through Bangladesh along the track and
has weakened into a depression on May 4, it has also caused a large amount of rainfall over Bangladesh and adjoining areas
of northeastern states of India (Assam, Meghalaya and Arunachal Pradesh). On the other hand, the storm surge information

from the Emergency Response Coordination Centre (ERCC) portal in association with JRC under NOAA/HWRF (Figure 7)
also depicts that the part of Purba Medinipur and South 24 Parganas district of West Bengal have experienced high-level
storm surge, varying in height from 2 to 3 metres during the storm event. The coastal tract of Puri, Baleshwar & the part of
Bhadrak district (Odisha) has faced moderate level storm surge (1−2 metres high) with the coastal part of Purba Medinipur
and South 24 Parganas (West Bengal). The rest of the South 24 Parganas district from the coastal tract of West Bengal and

Jagatsinghapur & Kendrapara districts from the coastal tract of Odisha have also experienced low-level storm surge during
cyclonic storm Fani. So, the heavy rainfall and powerful storm surge have jointly caused major damages due to inundation in
the coastal districts of Odisha and West Bengal.

The cyclone affected regions of Odisha and West Bengal have been divided into five equal width zones (15 km each) based
on the distance from the coast (Figure 7) for the more detailed explanation, but it doesn't signify that any inundation has not

occurred beyond that extent. The results derived from the analysis of the SAR images in GEE environment (Table 5) help to
understand the overall flood scenario over the major affected regions due to heavy rainfall and powerful storm surge. The
computed inundation statistics claim that Puri, one of the coastal district of Odisha has faced major flooding due to a large
amount of rainfall with a high level of storm surge during the storm event. Besides, the other parts of the first zone (0-15 km)
within Odisha have also experienced large inundation than other zones due to the heavy downpour and moderate to low-level

storm surge because the advancement of ESCS Fani has occurred towards the Gangetic West Bengal almost along the
coastal tract of Odisha. On the other hand, the coastal tract of West Bengal has faced a significant inundation for
experiencing high to low-level of storm surge with heavy rainfall. The statistics also claims that the highest amount of
inundation has occurred in the fourth zone (45-60 km) of West Bengal than other parts of the cyclone-affected regions
because the summer cyclone Fani has moved towards Bangladesh through this zone and has caused heavy rainstorm on May

3, 2019. Figure 8 helps to get a better view of the most affected regions due to inundation during cyclonic storm Fani over
Odisha and West Bengal.





### 5.3 Other damage statistics

If the flooding has been explained as the only consequence of the devastating tropical cyclone, the severity of the storm will be underestimated. The intensive rainfall, strong winds and storm surges cause several other damages during storm event
across the world, making the coastal people vulnerable to cyclone (Khalil, 1993; Wang and Xu, 2008; Li and Li, 2013; Ward et al., 2011; Done et al., 2018; Mori and Takemi, 2016). The deadly cyclonic storm Fani has also unleashed copious rainfall with the powerful windstorm that has gusted up to 205 kmph, leading to fatalities, complete destruction of kutcha houses, partial damage to buildings and others properties like roads, power sectors, health care centres, educational sector, embankment, etc. The trail of devastation caused by ESCS Fani in the large part of Odisha with West Bengal and Andhra
Pradesh has been explained briefly in Table 6.

### 6 Conclusion

The present study has focused on the historical background (1891-2019) of TCs over NIO with particular reference to ESCS Fani with meteorological variability, India's preparedness and awful aftermath. The discussion helps to understand that the probabilities of intensification of a cyclonic storm into a severe or more intense cyclone over the northern BoB and AS is
maximum during the monsoon season (June to September), differs from the intensification period of the cyclones that originate over the southern BoB and AS mainly during pre-monsoon (March to May) and post-monsoon (October to December) seasons. The recent work also confirms that the Utkal (Odisha), Bengal and Andhra coast receive higher frequency of TC than any other coastal tracts of India. Fani, one of the three worst cyclonic storms in last 150 years during pre-monsoon season has made landfall near Puri of Odisha coast and causes devastation over the extensive parts of mainly
Odisha and West Bengal, in spite of taking zero casualty approach by the local and national government. Generally, there are two possible ways of obtaining the complete resilience against such intense TCs, namely, the pre-storm preparedness plans understanding the risk based on early warning and also resilient infrastructure. India has already abled to achieve its first goal. IMD has made the work easier for the Indian government. But it is quite impossible to manage the second goal, resilient infrastructure for every citizen in a highly populous country like India. For that reason, the government has to build
sufficient cyclone and flood shelters so that evacuated people get protection during the storm event and the death toll can be reduced. The several state-level housing schemes and national housing schemes like Indira Awas Yojana (IAY) and Pradhan Mantri Awas Yojana (PMAY) for the economically weaker section can also cause the advancement towards a long-term solution to ensure full resilience against cyclonic storms slowly. So India can't get off immediately from the massive economic losses caused by TCs almost every year until the nation becomes successful to achieve the final goal with time.

**Data availability.** A wide range of datasets like the historical background of the cyclonic storm distribution data over NIO (1891-2019), Sentinel-1 SAR images, GPM IMERG Final Precipitation data, sustained wind speed & atmospheric pressure related data and storm surge height information from the Emergency Response Coordination Centre (ERCC) portal have





been used in the present study. Besides, several reports published on ESCS Fani by UNICEF (2019), OSDMA, UN, WB &
ADB (2019), IRC (2019) and Humanity Road (2019) have helped to gather sufficient preparedness and damage related
information. There is no related supplement for this paper.

**Acknowledgement.** This study is largely benefited from the Indian Meteorological Organization (IMD), Regional
Specialized Meteorological Centre (RSMC), Building Materials & Technology Promotion Council (BMTPC), Odisha State
Disaster Management Authority (OSDMA), World Bank (WB), Asian Development Bank (ADB), National Aeronautics and
Space Administration (NASA), European Space Agency (ESA), United Nations (UN) and Emergency Response
Coordination Centre (ERCC). The author wants to acknowledge Giovanni for allowing to visualize daily rainfall distribution
during the storm event and Google Earth Engine (GEE) for providing the opportunity to analyze and visualize the flooded
areas. Finally, I also wish to express my heartfelt gratitude to the anonymous reviewers and the Editors of Natural Hazards
and Earth System Sciences (NHESS) for their thoughtful and thorough reviews that improved the clarity of the manuscript.

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



Natural Hazards
and Earth System
Figure 1: (a) Map showing the genesis of cyclonic disturbances (CS and above) over the North Indian Ocean (NIO) from 1891 to 2018 by dividing the entire region into 2°30′ X 2°30′ grid based on Indian Meteorological Department (IMD) data source. (b) Seasonal (Pre-monsoon, Monsoon and Post-monsoon) distribution of cyclonic storms over the Arabian Sea and Bay of Bengal at the same time frame (1891-2018). The grid-wise data of NIO have been collected from the © Regional Specialized Meteorological Centre (RSMC) for Tropical Cyclones over North Indian Ocean, Indian Meteorological Department (IMD).


Figure 2: (a) Spatio-temporal distribution and trend analysis of tropical cyclones over the Bay of Bengal (BoB) and Arabian Sea (AS) for understanding the intensification rate of cyclonic storms to severe or more intensed cyclonic storms over: (b) BoB and (c) AS.




**Figure 3: Two windows as region of interest (roi) for extracting major inundated regions over West Bengal and Odisha during ESCS Fani by analyzing the Sentinel-1 SAR GRD data in © Google Earth Engine (GEE) environment.**





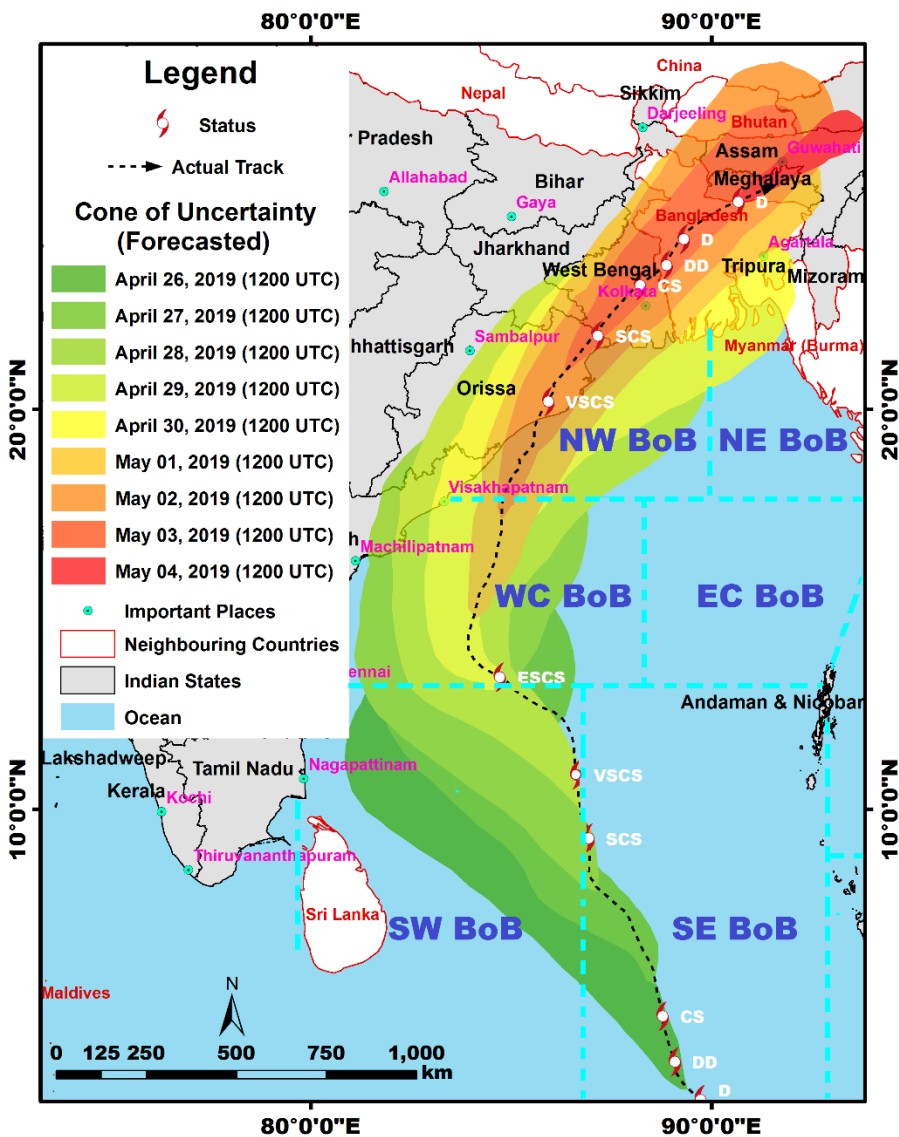

**Figure 4: Advancement of ESCS Fani over time (26 April to 04 May 2019) has been shown with forecasted Cone of Uncertainty and storm status, such as Depression (D), Deep Depression (DD), Cyclonic Storm (CS), Severe Cyclonic Storm (SCS), Very Severe Cyclonic Storm (VSCS) and Extremely Severe Cyclonic Storm (ESCS) from the report of © Regional Specialized Meteorological Centre (RSMC) for Tropical Cyclones over North Indian Ocean, Indian Meteorological Department (IMD). The Generic names like tropical cyclone or cyclonic storm is used to represent CS, SCS, VSCS and ESCS as a whole.**






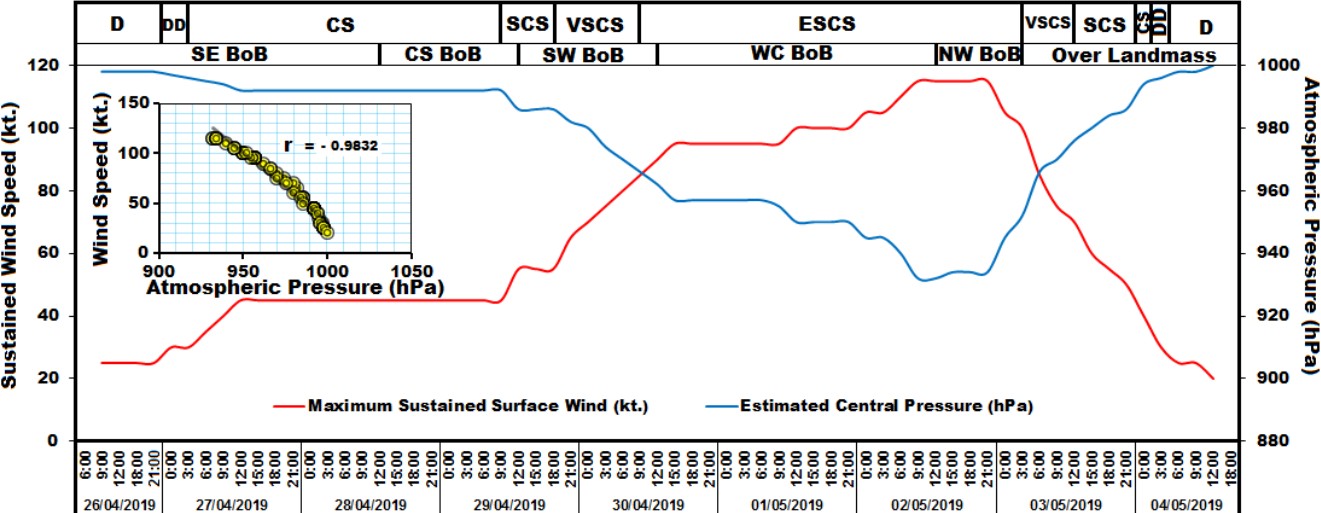

**Figure 5: Relationship between maximum sustained surface wind and estimated central pressure along the best track of cyclonic storm Fani with detailed status (CS, SCS, VSCS and ESCS) and track location.**

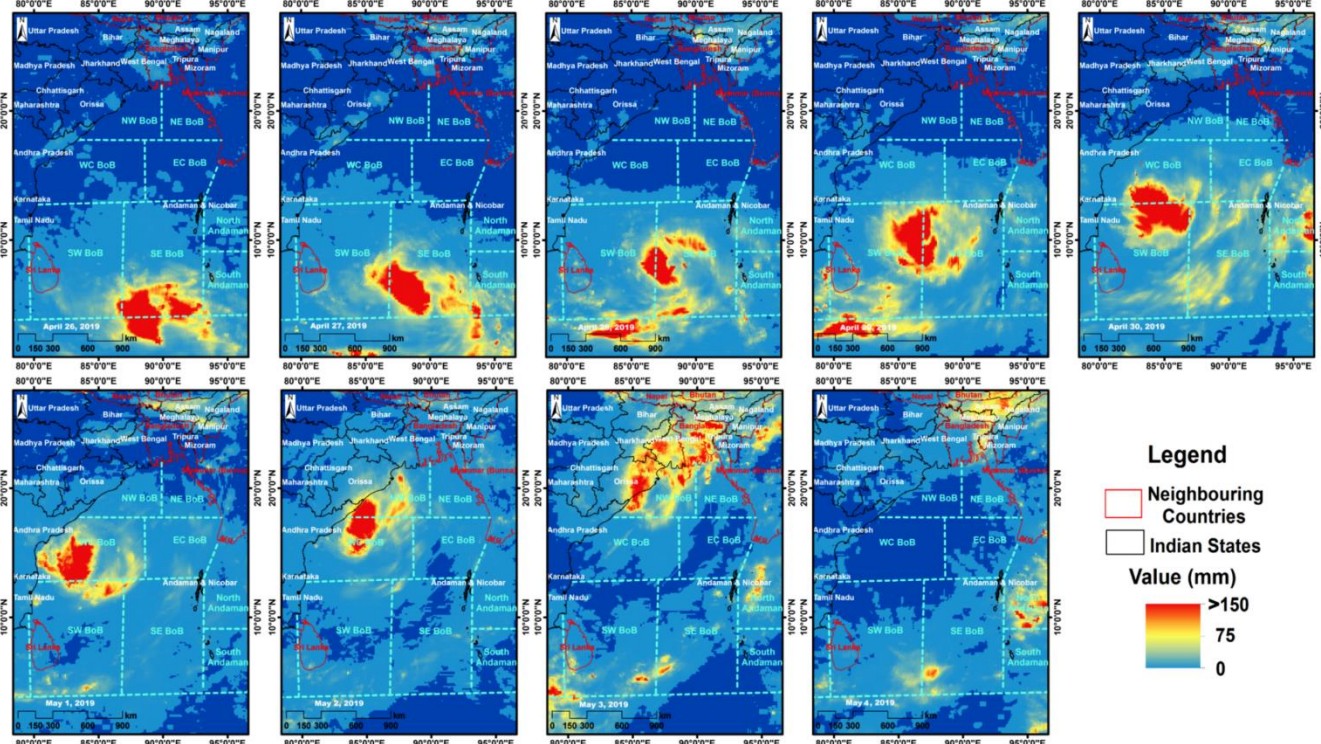

**Figure 6: Distribution of rainfall over the Bay of Bengal and its adjacent Indian subcontinent, using NASA's GPM IMERG Final images.**





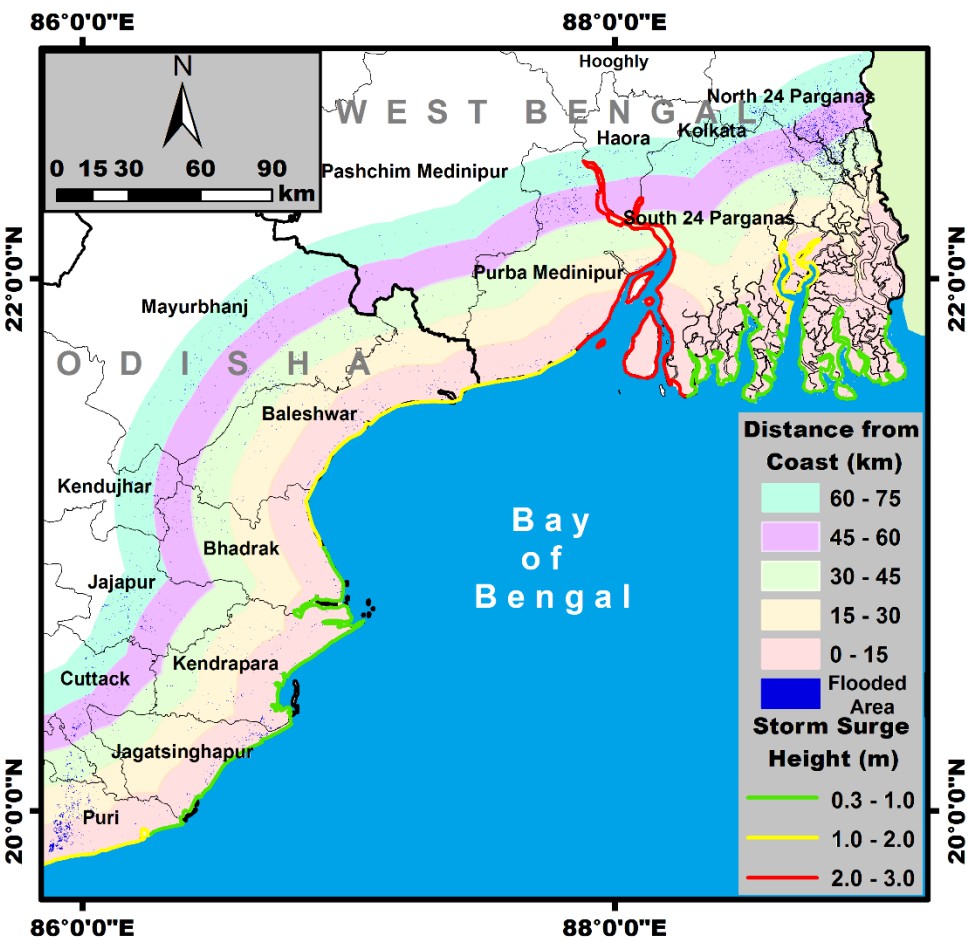


**Figure 7: Flood map of West Bengal and Odisha (based on distance from the coast) with storm surge height (metres above sea level) information from © Emergency Response Coordination Centre (ERCC) during ESCS Fani.**
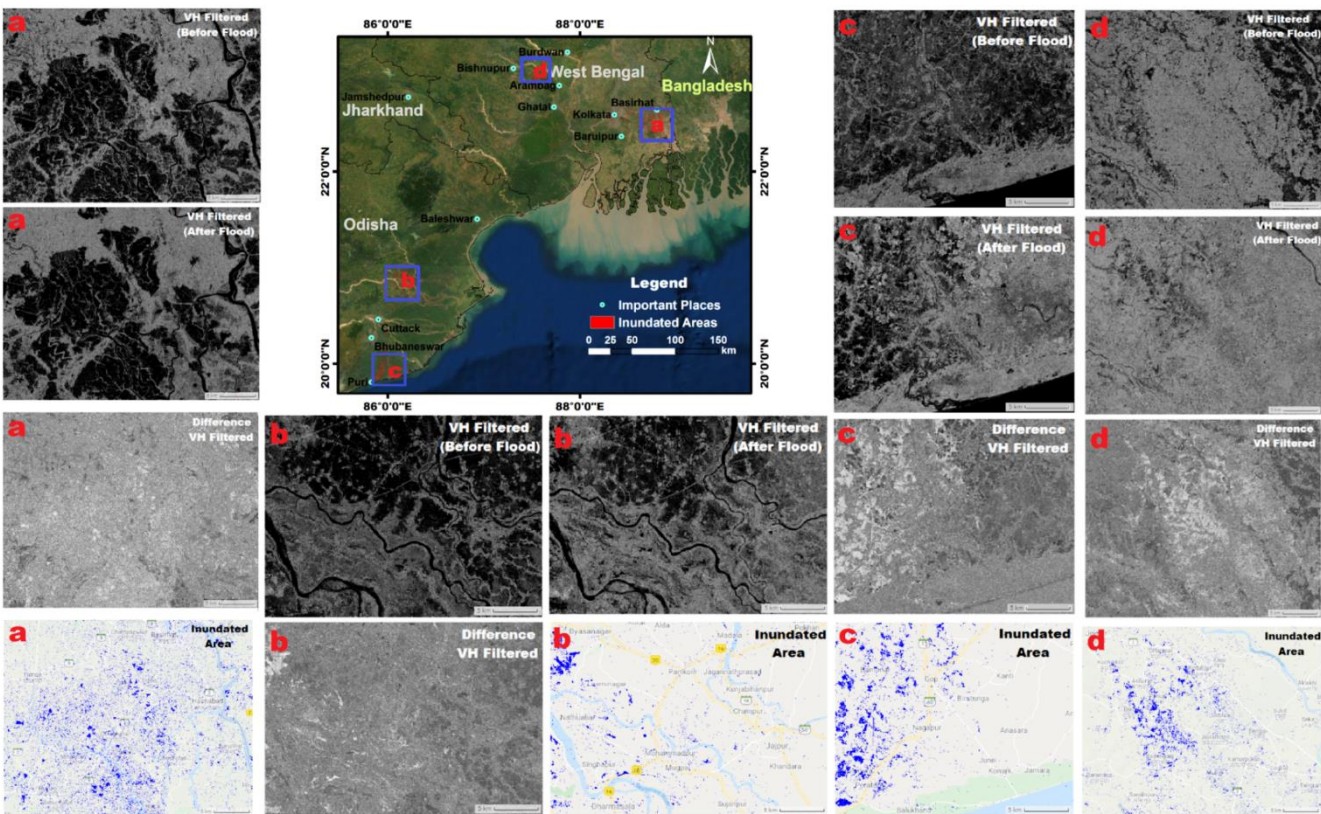

**Figure 8: Flood water identification by analyzing the Sentinel-1 SAR GRD: C-band (Polarisation: VH & VV) data, using © Google**
**Earth Engine (Base map sources: © Esri, DigitalGlobe, GeoEye, Earthstar Geographics, CNES/Airbus DS, USDA, USGS,**
**AeroGRID, IGN, and the GIS User Community). Severely affected regions of Odisha and West Bengal due to flooding that has**
**been caused by havoc rains and storm surges have been shown: (a) Portion of North 24 Parganas district near Basirhat of West**
**Bengal, (b) Part of Jajpur district near Dharmasala of Odisha, (c) Extensive part of Puri district of Odisha, and (d) Hooghly**
**district and its adjoining region of West Bengal. The blue patches represent flood water extent during ESCS Fani.**


**Table 1: Dataset used for the analysis and visualization different aspects of Very Severe Cyclone Fani.**

| Data | Imaging Date | Entity ID/ Usage | Data Source |
|------|------|------|------|
|  |  |  |  |





| | | | |
|---|---|---|---|
| COPERNICUS/ Sentinel-1 SAR GRD: C-band (Polarization: VH & VV) | Before Cyclone Fani (15 Feb - 21 Feb, 2019) | S1A_IW_GRDH_1SDV_20190221T000435_20190221T000500_ 026020_02E673_C02D<br>S1A_IW_GRDH_1SDV_20190221T000500_20190221T000528_ 026020_02E673_4CED<br>S1A_IW_GRDH_1SDV_20190221T000410_20190221T000435_ 026020_02E673_648B<br>S1A_IW_GRDH_1SDV_20190215T235603_20190215T235628_ 025947_02E3DD_CE8B<br>S1A_IW_GRDH_1SDV_20190215T235628_20190215T235653_ 025947_02E3DD_B50F<br>S1B_IW_GRDH_1SDV_20190215T000355_20190215T000424_ 014949_01BE99_45F5 | https://sentinel.es a.int/web/sentine l/user-guides/sentinel-1-sar/ |
| | After Cyclone Fani (04 May, 2019) | S1A_IW_GRDH_1SDV_20190504T000447_20190504T000512_ 027070_030CB5_31EF<br>S1A_IW_GRDH_1SDV_20190504T000512_20190504T000537_ 027070_030CB5_CDC5<br>S1A_IW_GRDH_1SDV_20190504T000357_20190504T000422_ 027070_030CB5_EF09<br>S1A_IW_GRDH_1SDV_20190504T000422_20190504T000447_ 027070_030CB5_863C<br>S1B_IW_GRDH_1SDV_20190504T235513_20190504T235542_ 016101_01E49E_F422<br>S1B_IW_GRDH_1SDV_20190504T235542_20190504T235616_ 016101_01E49E_4F22 | |
| GPM IMERG Final Precipitation L3 1 day 0.1 degree x 0.1 degree V06 (GPM_3IMERG DF) | 26 April - 05 May, 2020 | 3B-DAY.MS.MRG.3IMERG.20190426-S000000-E235959.V06<br>3B-DAY.MS.MRG.3IMERG.20190427-S000000-E235959.V06<br>3B-DAY.MS.MRG.3IMERG.20190428-S000000-E235959.V06<br>3B-DAY.MS.MRG.3IMERG.20190429-S000000-E235959.V06<br>3B-DAY.MS.MRG.3IMERG.20190430-S000000-E235959.V06<br>3B-DAY.MS.MRG.3IMERG.20190501-S000000-E235959.V06<br>3B-DAY.MS.MRG.3IMERG.20190502-S000000-E235959.V06<br>3B-DAY.MS.MRG.3IMERG.20190503-S000000-E235959.V06<br>3B-DAY.MS.MRG.3IMERG.20190504-S000000-E235959.V06 | https://disc.gsfc. nasa.gov/datasets /GPM_3IMERG DF_06/summary |





**Table 2: Accuracy assessment table for measuring the precision of extracted flooded area from Sentinel-1 SAR images during ESCS Fani.**

| Classification Data | Accuracy assessment value (%) | | | | | | | | | | | |
|---|---|---|---|---|---|---|---|---|---|---|---|---|
| | Odisha and surroundings (Window 2) | | | | | | West Bengal and surroundings (Window 1) | | | | | |
| | VH polarization | | | VV polarization | | | VH polarization | | | VV polarization | | |
| | Producer's Accuracy | User's Accuracy | Overall | Producer's Accuracy | User's Accuracy | Overall | Producer's Accuracy | User's Accuracy | Overall | Producer's Accuracy | User's Accuracy | Overall |
| Water bodies | 100 | 100 | 92.92 | 100 | 100 | 92.50 | 100 | 100 | 95.42 | 100 | 100 | 93.75 |
| Flooded lands | 88.88 | 90 | | 89.74 | 87.50 | | 100 | 86.25 | | 98.51 | 82.50 | |
| Non-flooded lands | 89.87 | 88.75 | | 87.80 | 90 | | 88.89 | 100 | | 84.95 | 98.75 | |


**Table 3: Evaluation of research articles and reports on Extremely Severe Cyclonic Storm (ESCS) Fani.**

| Type | Entity | Focused Areas of the study | | | | | | | | | |
|---|---|---|---|---|---|---|---|---|---|---|---|
| | | Synoptic history | Model-based prediction and forecast | Storm surge information | Damage Statistics | Impact on global atmosphere | Impact on different land use/ land cover | Coastal impacts | Ecological impacts | Morpho-dynamics impacts | Impact on power sector |
| Research Articles | Acharyya et al. (2020) | | | | | | | ✓ | ✓ | ✓ | |
| | Kumar et al. (2020) | ✓ | | | | | ✓ | | | | |
| | Liu and Zhu (2020) | | | | | ✓ | | | | | |
| | Mohanty et al. (2020a) | ✓ | ✓ | | ✓ | | | | | | |
| | Mohanty et al. (2020b) | | | | | | | | | | ✓ |
| Reports | Humanity Road (2019) | | | | ✓ | | | ✓ | | | ✓ |
| | IRC (2019) | | | ✓ | ✓ | | | | | | |
| | OSDMA, UN, WB & ADB (2019) | | | ✓ | ✓ | | ✓ | ✓ | ✓ | | ✓ |
| | UNICEF (2019) | | | ✓ | ✓ | | | | | | |

**Table 4: Best track details for Very Severe Cyclonic Storm Fani, 26 April - 4 May 2019.**



| Date/Time | | Lat. (°N) | Long. (°E) | Estimated Central Pressure (hPa) | Average Wind Speed | | Status | Remarks |
|---|---|---|---|---|---|---|---|---|
| UTC | IST | | | | kt. | km. | | |
| 26/06:00 | 26/11:30 | 3 | 89.4 | 998 | 25 | 46.25 | D | Over South-east Bay of Bengal (SE BoB) |
| 27/00:00 | 27/05:30 | 4.5 | 88.8 | 997 | 30 | 55.50 | DD | |
| 27/03:00 | 27/08:30 | 4.9 | 88.7 | 996 | 30 | 55.50 | DD | |
| 27/06:00 | 27/11:30 | 5.2 | 88.6 | 995 | 35 | 64.75 | CS | |
| 27/09:00 | 27/14:30 | 5.4 | 88.5 | 994 | 40 | 74.00 | CS | |
| 27/12:00 | 27/17:30 | 5.9 | 88.5 | 992 | 45 | 83.25 | CS | |
| 28/12:00 | 28/17:30 | 8.2 | 87 | 992 | 45 | 83.25 | CS | |
| 29/12:00 | 29/17:30 | 10.1 | 86.7 | 986 | 55 | 101.75 | SCS | Over Central Southern Bay of Bengal (CS BoB) |
| 29/21:00 | 30/02:30 | 11.1 | 86.5 | 982 | 65 | 120.25 | VSCS | |
| 30/00:00 | 30/05:30 | 11.7 | 86.5 | 980 | 70 | 129.50 | VSCS | Over South-west Bay of Bengal (SW BoB) |
| 30/03:00 | 30/08:30 | 12.3 | 86.2 | 974 | 75 | 138.75 | VSCS | |
| 30/06:00 | 30/11:30 | 12.6 | 85.7 | 970 | 80 | 148.00 | VSCS | |
| 30/09:00 | 30/14:30 | 13 | 85.3 | 966 | 85 | 157.25 | VSCS | |
| 30/12:00 | 30/17:30 | 13.3 | 84.7 | 962 | 90 | 166.50 | ESCS | |
| 30/15:00 | 30/20:30 | 13.4 | 84.5 | 957 | 95 | 175.75 | ESCS | Over West-central Bay of Bengal (WC BoB) |
| 01/09:00 | 01/14:30 | 14.5 | 84.1 | 955 | 95 | 175.75 | ESCS | |
| 01/12:00 | 01/17:30 | 14.9 | 84.1 | 950 | 100 | 185.00 | ESCS | |
| 02/00:00 | 02/05:30 | 15.9 | 84.5 | 945 | 105 | 194.25 | ESCS | |
| 02/06:00 | 02/11:30 | 16.7 | 84.8 | 940 | 110 | 203.50 | ESCS | |
| 02/09:00 | 02/14:30 | 17.1 | 84.8 | 932 | 115 | 212.75 | ESCS | |
| 02/15:00 | 02/20:30 | 17.8 | 84.9 | 934 | 115 | 212.75 | ESCS | Over North-west Bay of Bengal (NW BoB) |
| 03/03:00 | 03/08:30 | 19.6 | 85.7 | 952 | 100 | 185.00 | ESCS | |
| 03/06:00 | 03/11:30 | 20.2 | 85.9 | 966 | 85 | 157.25 | VSCS | Crossed Odisha coast close to Puri (near lat. 19.75N and Long. 85.70E ) between 0230 to 0430 UTC on 3 May 2019 (Over Odisha & West |
| 03/09:00 | 03/14:30 | 20.6 | 86 | 970 | 75 | 138.75 | VSCS | |
| 03/12:00 | 03/17:30 | 21.1 | 86.5 | 976 | 70 | 129.50 | VSCS | |
| 03/15:00 | 03/20:30 | 21.5 | 86.7 | 980 | 60 | 111.00 | SCS | |
| 03/18:00 | 03/23:30 | 21.9 | 87.1 | 984 | 55 | 101.75 | SCS | |
| 03/21:00 | 04/02:30 | 22.5 | 87.9 | 986 | 50 | 92.50 | SCS | |



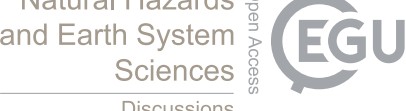

| 04/00:00 | 04/05:30 | 23.1 | 88.2 | 994 | 40 | 74.00 | CS | Bengal) |
| 04/03:00 | 04/08:30 | 23.6 | 88.8 | 996 | 30 | 55.50 | DD | Over Bangladesh |
| 04/06:00 | 04/11:30 | 24.3 | 89.3 | 998 | 25 | 46.25 | D | |
| 04/12:00 | 04/17:30 | 25.2 | 90.7 | 1000 | 20 | 37.00 | D | Over central Assam & neighbourhood |

Note: Several abbreviations have been used in this table as status of the storm as IMD scale, such as Depression (D), Deep

Depression (DD), Cyclonic Storm (CS), Severe Cyclonic Storm (SCS), Very Severe Cyclonic Storm (VSCS) and Extremely

Severe Cyclonic Storm (ESCS).

**Table 5: Inundation statistics of Odisha and West Bengal during Extremely Severe Cyclonic Storm (ESCS) Fani.**

| Distance from coast (km) | Inundated Area (Sq. km) | |
| --- | --- | --- |
| | Odisha | West Bengal |
| 0 – 15 | 47.99 | 50.41 |
| 15 – 30 | 20.03 | 19.22 |
| 30 – 45 | 18.05 | 39.49 |
| 45 – 60 | 20.67 | 63.51 |
| 60 – 75 | 23.07 | 30.67 |

**Table 6: Damage caused by the Very Severe Cyclonic Storm Fani in Odisha, West Bengal and Andhra Pradesh of India (*Source*: UNICEF, 2019; Odisha State Disaster Management Authority, United Nations, World Bank and ADB 2019).**

| **Total Damage** across Odisha, West Bengal and Andhra Pradesh | People affected (million) | | 28 |
| --- | --- | --- | --- |
| | Districts affected | | 24 |
| | Deaths reported (all from Odisha) | | 64 |
| | Building affected | | 4610 |
| | Livestock Casualty (million) | | 3.73 |
| **Affected or Damage** in Odisha | People affected (million) | | 16.5 |
| | Districts affected | | 14 |
| | Villages affected | | 18,388 |
| | Deaths reported | | 64 |
| | Perennial crops damage (sq. km) | | 197 |
| | Total estimated costs in USD (billion) | | 4.18 |
| | Casualties of poultry birds (million) | | 5.4 |
| | Fisheries | Traditional fishermen (million) | 0.15 |





| | | Traditional marine fishing boats | 6,416 |
|---|---|---|---|
| | | Fishing ponds | 2,524 |
| | Educational sector | Primary and Secondary schools | 5,735 |
| | | Colleges (Govt. & Govt. aided) | 102 |
| | | Universities | 5 |
| | Health sector | Public Health Facilities | 1,031 |
| | | Anganwadi centres | 2,513 |
| | Houses damaged | Rural | 295,703 |
| | | Urban | 66,040 |
| | Power sector | High tension poles | 200 |
| | | Distribution transformers | 11,077 |
| Length of damaged embankments (km) | | | 22.67 |
| Length of damaged roads (km) | | | 11,763 |
| Trees damaged (million) | | | 2.19 |