# Peer review of "Analytical Study of North Indian Oceanic Cyclonic Disturbances with Special Reference to Extremely Severe Cyclonic Storm Fani: Meteorological Variability, India's Preparedness with Terrible Aftermath"

_Natural Hazards and Earth System Sciences, 2020_

## Referee Comment (RC1) · Yuan-Jung TSAI (Referee) · 16 Oct 2020

The correlation between cyclones and flooding disasters was purposed in this manuscript, but most of the content was about the variation in time and special distribution of cyclones, and there was little description of disasters.

The main factor of the correlation was discussed with the effect of the cyclones, but flooding disasters are not only related to the rainfall, but also surface elevation, water

system distribution, urban development, etc. However, in this article, only the affected area (distance) of the cyclones is considered, and other factors should be considered.

Base on the result, only possible to evacuation plan were proposed as the government actions. In this case, the cost might be very high with the dense crowd and conservative hazard mapping.

---

## Referee Comment (RC2) · Anonymous Referee #2 · 16 Oct 2020

This paper focused on the historical background of tropical cyclones over North Indian Ocean with particular reference to cyclonic storm Fani. The cyclonic disturbances from 1891 to 2018 was detailed described, and the developing trend was simply plotted. The flooded area of storm Fani was mapped by using Sentinel-1 SAR dataset. The discussion mainly focused on India's preparedness, inundation detail and some damage statistics. After reading this paper, to be frank I can't find significant academic contributions of this study. It looks more like a general report, the in-depth scientific

analysis or discussion obviously is insufficient.

---

## Author Comment (AC1) · 30 Oct 2020

Respected Sir, I want to thank for your constructive comments on my manuscript. As I have focused on the analytical study of North Indian Oceanic cyclonic disturbances, I have included the long-term (of last 150 years) annual and seasonal distribution of tropical cyclones over the Arabian Sea (AS) and Bay of Bengal (BoB). It would help us to understand the most affected regions by cyclones over the North Indian Ocean

(NIO). On the other hand, to give a more specific example on the terrible aftermath over the eastern coast, I have selected one of the recent and rarest cyclonic storms Fani with its detailed meteorological variability with overall India's preparedness. It would also help to understand how the local and national government have tackled this deadly cyclone and what could be done to minimize the casualty to face that kind of monstrous cyclone in near future. That is the main reason behind the selection of the only one extreme event (Fani) with detailed scenario in the present manuscript. Firstly, you have stated that "The main factor of the correlation was discussed with the effect of the cyclones, but flooding disasters are not only related to the rainfall, but also surface elevation, water system distribution, urban development, etc. However, in this article, only the affected area (distance) of the cyclones is considered, and other factors should be considered". I am completely same-minded to you. For that reason, I have tried to include one paragraph at the end of the "5.2 Inundation detail" section after line number 216 to correlate the inundation with surface elevation, water system distribution, urban development, etc. That paragraph with a new figure has been given below: The maximum flood extent of the study area can be found up to 30 meters altitude from the mean sea level (MSL). The calculated areal extension of flooding (up to 75 km inside from the coast) based on surface elevation also helps to establish the same fact (Figure 9). Nearly 69.96% of the total inundated area is extended up to 10 meters altitude from MSL. On the other hand, 21.26% of the flood has occurred between 10-20 meters and only 7.96% of the total flooded area has been found in between 20-30 meters altitudinal extension. As the ESCS Fani has made the landfall near Puri of Odisha, the interfluve region of Bhargabi and Kaathajodi River has experienced severe flooding. The heavy rainfall has caused flooding in the deltaic portion of Mahanadi River for having lower altitude (< 10 meters from MSL) and the presence of distributaries and innumerable rivulets. Besides, the interfluve regions of Haldi-Rupnarayan, Silabati-Dwarakeswar and Ganges deltaic portion of West Bengal has also experienced flooding. No urban area except Puri has been affected due to flooding during this cyclonic event, but the large infrastructural destruction has been

made by cyclonic storm Fani due to high velocity of gusty wind in almost every small to large urban centres of the coastal districts.

Secondly, you have also stated that "Base on the result, only possible to evacuation plan were proposed as the government actions. In this case, the cost might be very high with the dense crowd and conservative hazard mapping". I also completely agree with you. As that portion of the country are very densely populated and several billions of people inhabit in this place, taking a particular preparedness plan is very challenging for the high and very densely populated place. For that reason, I have stated one short-term and one long-term plan that would help to tackle that kind of disastrous natural hazards: [1] Evacuation from the track of the cyclone (short-term plan): We need to build sufficient numbers of cyclone and flood shelters so that evacuated people get protection during the storm event and the death toll can be reduced. [2] Housing schemes (long-term plan): The several state-level housing schemes and national housing schemes like Indira Awas Yojana (IAY) and Pradhan Mantri Awas Yojana (PMAY) for the economically weaker section can also cause the advancement towards a long-term solution to ensure full resilience against cyclonic storms with time. Hopefully, I have been able to meet your queries.
* * *
**Fig. 1.** Figure 9: Major drainage system of the study area on NASA's © Shuttle Radar Topography Mission (SRTM) Digital Elevation Models (DEM) with the small to large affected urban centres.

---

## Author Comment (AC2) · 30 Oct 2020

Respected Sir, I want to thank for your constructive comments on my manuscript. As I have focused on the analytical study of North Indian Oceanic cyclonic disturbances, I have included the long-term (of last 150 years) annual and seasonal distribution of tropical cyclones over the Arabian Sea (AS) and Bay of Bengal (BoB). It would also help us to understand the most affected regions by cyclones over the North Indian

Ocean (NIO). On the other hand, to give a more specific example on the terrible aftermath over the eastern coast, I have selected one of the recent and rarest cyclonic storms Fani with its detailed meteorological variability with overall India's preparedness. It would also help to understand how the local and national government have tackled this deadly cyclone and what could be done to minimize the casualty to face that kind of monstrous cyclone in near future. That is the main reason behind the selection of the only one extreme event (Fani) with detailed scenario in the present manuscript. Firstly, you have stated that "This paper focused on the historical background of tropical cyclones over North Indian Ocean with particular reference to cyclonic storm Fani. The cyclonic disturbances from 1891 to 2018 was detailed described, and the developing trend was simply plotted. The flooded area of storm Fani was mapped by using Sentinel-1 SAR dataset. The discussion mainly focused on India's preparedness, inundation detail and some damage statistics. After reading this paper, to be frank I can't find significant academic contributions of this study. It looks more like a general report, the in-depth scientific analysis or discussion obviously is insufficient". I have tried my best to justify my manuscript. This manuscript could be the first one which includes several parameters at the same time: [1] The historical background and trend analysis of tropical cyclones (TCs) over NIO. [2] State-level and national- level preparedness to fight against the deadly cyclonic storm like Fani. [3] Visualization of the flood-affected area by analyzing Sentinel-1 SAR dataset in GEE environment. [4] Zone wish quantification of inundated areas to understand the flood situation with other damage statistics. [5] Highlights the reasons behind recognition from WMO and UN. Besides, I have tried to include one more paragraph at the end of the "5.2 Inundation detail" section after line number 216 to correlate the inundation with surface elevation, water system distribution, urban development, etc. That paragraph with a new figure has been given below: The maximum flood extent of the study area can be found up to 30 meters altitude from the mean sea level (MSL). The calculated areal extension of flooding (up to 75 km inside from the coast) based on surface elevation also helps to establish the same fact (Figure 9). Nearly 69.96% of the total inundated area is

extended up to 10 meters altitude from MSL. On the other hand, 21.26% of the flood has occurred between 10-20 meters and only 7.96% of the total flooded area has been found in between 20-30 meters altitudinal extension. As the ESCS Fani has made the landfall near Puri of Odisha, the interfluve region of Bhargabi and Kaathajodi River has experienced severe flooding. The heavy rainfall has caused flooding in the deltaic portion of Mahanadi River for having lower altitude (< 10 meters from MSL) and the presence of distributaries and innumerable rivulets. Besides, the interfluve regions of Haldi-Rupnarayan, Silabati-Dwarakeswar and Ganges deltaic portion of West Bengal has also experienced flooding. No urban area except Puri has been affected due to flooding during this cyclonic event, but the large infrastructural destruction has been made by cyclonic storm Fani due to high velocity of gusty wind in almost every small to large urban centres of the coastal districts. Hopefully, it would be enough to meet your queries.
* * *
[Figure]

**Legend**

⎯⎯ Rivers

■ Flooded Area

• Urban Centers

**Elevation (m.)**

| | |
|---|---|
| | < 10 |
| | 10.01 - 20 |
| | 20.01 - 30 |
| | 30.01 - 40 |
| | 40.01 - 50 |
| | 50.01 - 60 |
| | 60.01 - 70 |
| | 70.01 - 80 |
| | 80.01 - 90 |
| | 90.01 - 100 |
| | 100.01 - 110 |
| | 110.01 - 120 |
| | 120.01 - 130 |
| | 130.01 - 140 |
| | 140.01 - 150 |
| | 150.01 - 160 |
| | 160.01 - 170 |
| | 170.01 - 180 |
| | 180.01 - 190 |
| | > 190 |

**Fig. 1.** Figure 9: Major drainage system of the study area on NASA's © Shuttle Radar Topography Mission (SRTM) Digital Elevation Models (DEM) with the small to large affected urban centres.

---

## Author Comment (AC3) · 13 Dec 2020

Respected Sir, I want to thank for your constructive comments on my manuscript. As I have focused on the analytical study of North Indian Oceanic cyclonic disturbances, I have included the long-term (of last 150 years) annual and seasonal distribution of tropical cyclones over the Arabian Sea (AS) and Bay of Bengal (BoB). It would help us to understand the most affected regions by cyclones over the North Indian Ocean

(NIO). On the other hand, to give a more specific example on the terrible aftermath over the eastern coast, I have selected one of the recent and rarest cyclonic storms Fani with its detailed meteorological variability with overall India's preparedness. It would also help to understand how the local and national government have tackled this deadly cyclone and what could be done to minimize the casualty to face that kind of monstrous cyclone in near future. That is the main reason behind the selection of the only one extreme event (Fani) with detailed scenario in the present manuscript.

Firstly, you have stated that "The main factor of the correlation was discussed with the effect of the cyclones, but flooding disasters are not only related to the rainfall, but also surface elevation, water system distribution, urban development, etc. However, in this article, only the affected area (distance) of the cyclones is considered, and other factors should be considered". I am completely same-minded to you. For that reason, I have tried to include one paragraph at the end of the "5.2 Inundation detail" section after line number 216 to correlate the inundation with surface elevation, water system distribution, urban development, etc. In the track change version of that manuscript, I have also made these changes which are little bit different from the previous reply. That paragraph with a new figure has been given below:

"On the other hand, the more detailed analysis helps to determine that the maximum flood extent of the study area can be found up to 40 meters altitude from the mean sea level (MSL) in the lower catchment areas (floodplain region) and also along the low lying areas of the river valleys. The calculated areal extension of flooding (up to 75 km inside from the coast) based on surface elevation also helps to establish the same fact (Figure 9). Nearly 69.96% of the total inundated area is extended up to 20 meters altitude from MSL. On the other hand, 21.26% of the flood has occurred between 20-30 meters and only 7.96% of the total flooded area has been found in between 30-40 meters altitudinal extension. As the ESCS Fani has made the landfall near Puri of Odisha, the interfluve region of Bhargabi and Kaathajodi River has experienced severe flooding. The heavy rainfall has caused flooding in the deltaic portion of Mahanadi

River for having lower altitude (< 10 meters from MSL) and the presence of distributaries and innumerable rivulets. Besides, the interfluve regions of Haldi-Rupnarayan, Silabati-Dwarakeswar and Ganges deltaic portion of West Bengal has also experienced flooding. No urban area except Puri has been affected due to flooding during this cyclonic event, but the large infrastructural destruction has been made by cyclonic storm Fani due to high velocity of gusty wind in almost every large to small urban centres of the coastal districts. The other damage history excluding flooding has been discussed in the next section below."

Secondly, you have also stated that "Base on the result, only possible to evacuation plan were proposed as the government actions. In this case, the cost might be very high with the dense crowd and conservative hazard mapping". I also completely agree with you. As that portion of the country are very densely populated and several billions of people inhabit in this place, taking a particular preparedness plan is very challenging for the high and very densely populated place. For that reason, I have stated one short-term and one long-term plan that would help to tackle that kind of disastrous natural hazards:

[1] Evacuation from the track of the cyclone (short-term plan): We need to build sufficient numbers of cyclone and flood shelters so that evacuated people get protection during the storm event and the death toll can be reduced.

[2] Housing schemes (long-term plan): The several state-level housing schemes and national housing schemes like Indira Awas Yojana (IAY) and Pradhan Mantri Awas Yojana (PMAY) for the economically weaker section can also cause the advancement towards a long-term solution to ensure full resilience against cyclonic storms with time.

Hopefully, I have been able to meet your queries.

Please also note the supplement to this comment:
https://nhess.copernicus.org/preprints/nhess-2020-287/nhess-2020-287-AC3-

supplement.pdf

**Fig. 1.** Figure 9: Large to small urban centres which are affected by Fani with major drainage system of the study area on NASA's © SRTM DEM.